# Collective Model Intelligence Requires Compatible Specialization

**Jyothish Pari**[*]
Massachusetts Institute of Technology
Cambridge, MA, USA

**Samy Jelassi**
Harvard University
Cambridge, MA, USA

**Pulkit Agrawal**
Massachusetts Institute of Technology
Cambridge, MA, USA

## Abstract

In this work, we explore the limitations of combining models by averaging intermediate features, referred to as *model merging*, and propose a new direction for achieving collective model intelligence through what we call *compatible specialization*. Current methods for model merging, such as parameter and feature averaging, struggle to effectively combine specialized models due to representational divergence during fine-tuning. As models specialize to their individual domains, their internal feature representations become increasingly incompatible, leading to poor performance when attempting to merge them for new tasks. We analyze this phenomenon using centered kernel alignment (CKA) and show that as models specialize, the similarity in their feature space structure diminishes, hindering their capacity for collective use. To address these challenges, we investigate routing-based merging strategies, which offer more flexible methods for combining specialized models by dynamically routing across different layers. This allows us to improve on existing methods by combining features from multiple layers rather than relying on fixed, layer-wise combinations. However, we find that these approaches still face limitations when layers within models are representationally incompatible. Our findings highlight the importance of designing new approaches for model merging that operate on well-defined input and output spaces, similar to how humans communicate through language rather than intermediate neural activations.

## 1 Introduction

Machine learning has recently seen an explosion of models trained for diverse tasks and data and made readily available on platforms such as Hugging Face. Given a new task, a question faced by practitioners is whether it is possible to reuse and combine information across existing models or whether a base foundation model should be fine-tuned for their application. We hypothesize that if a new task can be solved by combining existing models – such a collective model may outperform a model obtained by fine-tuning a base foundation model. We refer to the process of leveraging and combining multiple models to solve a target task as *model merging* (Raffel, 2023; Ha & Tang, 2022; Ferreira et al., 2024).

The idea of collectively utilizing the intelligence of individual entities is commonplace in nature – organisms often come together to form collectives—bee colonies, whale pods, and human societies. Coordination between individuals (or models) with potentially specialized roles (or functions) can enhance the capabilities of the whole (Seeley, 2009; Sumpter, 2010). Imagine we need to solve a problem related to disease modeling. Instead of training one person to be an expert in two fields, we may recruit two specialized experts – a mathematician

---

[*]Correspondence to: Jyothish Pari `<jyop@mit.edu>`

and a biologist – who can work together to solve the given problem. Effective collaboration requires the two experts to share a common language. Regardless of their skills, if one speaks only Hindi and the other speaks only German, their collaboration will be limited. Thus, effective collaboration requires entities that are specialized **and** can communicate in a shared language – a phenomenon we term *compatible specialization.*

We investigate state-of-the-art methods for *model merging* from the lens of *compatible specialization* and find that current approaches cannot effectively combine specialized models. A prominent way to merge models is to pool features of the input predicted by different models – we refer to this paradigm as representation-based or feature averaging. This includes techniques such as parameter averaging (Wortsman et al., 2022; Ilharco et al., 2022; Matena & Raffel, 2022; Yadav et al., 2024b; Yu et al., 2024) or computing a weighted average of the individual model features. (Ferreira et al., 2024; Fedus et al., 2022; Shazeer et al., 2017; Jiang et al., 2024a; Yadav et al., 2024a; Sukhbaatar et al., 2024; Tang et al., 2024). However, experiments show that feature averaging fails to achieve compatible specialization and is less effective than directly fine-tuning a base model for the target task.

To illustrate our perspective, consider this setup: we finetune a math model and a coding model from a common base foundation model, aiming to merge them for a new task that requires both math and coding skills—such as writing code to solve a math word problem. We now outline two key factors that prevent compatible specialization:

**1. Representations diverge during fine-tuning.**

To measure how "compatible" the math and coding models are while fine-tuning, we use a representational divergence metric, $D$. Our experiments reveal a critical point, $t$, where the dynamics of merging these models shift. From the start of fine-tuning up to $t$, as the math and coding models become more specialized in their respective domains, $D$ increases, yet merging still improves performance on tasks that require both math and coding skills, such as writing code to solve a math word problem. However, beyond $t$, further increases in $D$ lead to diminishing returns: merging yields fewer gains, and performance eventually degrades. This critical point $t$ marks the threshold where specialization begins to interfere with compatibility. Thus, merging the math and coding models is effective only when they haven't become too specialized.

**2. Layers become representationally incompatible.**

Traditionally, merging is done by fine-tuning models from the same base model, where features from the same depth in each model are combined (Zheng et al., 2024; Jiang et al., 2024b; Lange et al., 2022). We refer to layers at the same depth across different models as corresponding layers. However, a more flexible approach would allow for the combination of features from layers at different depths across multiple models, as layers at different depths may represent distinct functional computations. Unfortunately, the aforementioned trade-off between specialization and mergeability is mirrored here. As layers become more specialized, especially when they are positioned further apart in depth, representational divergence increases, limiting their compatibility for merging. Therefore, we cannot achieve compatible specialization across layers.

In essence, for model merging to be effective, models must not only be specialized but also able to communicate their knowledge in a way that facilitates collaboration rather than relying on representational alignment across layers. We argue that representations across finetuned models will remain fundamentally incompatible, even if they originate from a common base model. Therefore, model merging should not be approached as simply combining intermediate representations. Instead, it should focus on enabling models to exchange information, akin to how humans collaborate through language rather than combining their brain activity. This shift in perspective highlights the critical need to develop methods that support *compatible specialization.* While this paper does not propose a specific solution for achieving compatible specialization, it aims to identify this need and demonstrate that current model merging techniques fall short of it. We offer insights into how future work can move toward realizing true compatible specialization.

## 2 Representational Similarity Degrades over Time

> How does representational compatibility across specialized models relate to the merging performance?

As models become increasingly specialized, their internal representations tend to shift towards being more task-specific and less generalizable. We show this is Figure 1, where despite the task specific models become more capable at math and coding respectively, the merging performance on the adaptation task that requires math and coding abilities steadily degrades after initially improving. This posses a trade off where we need to balance specialization with representational compatibility. This is also clearly shown in the middle and right sub-figures in Figure 1 where we can see a "U" shape curve when plotting representational similarity vs merging performance. Therefore, from the start of fine-tuning, as the models acquire specializations, the merging performance improves. However, it appears after a critical step, the representations diverge to the point where further specialization does not contribute to merging performance.

It is import to note that we measure the representational similarity on the adaptation dataset that requires math and coding skills as well as the pretraining dataset OpenWebText (Gokaslan & Cohen, 2019). Having diverging representations on the pretraining datasets implies that the models have less compatibility. To mitigate this we try injecting the pretraining data into the fine-tuning process to enforce more compatibility, but we did not improve merging performance (see, Appendix A.8). This warrants the need to establish representational structure and compatibility which can mitigate the existence of the aforementioned trade off.

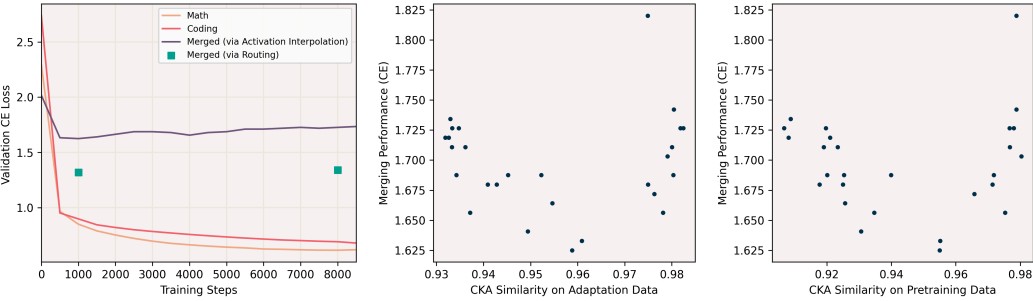

Figure 1: (Left) Validation cross-entropy loss (CE Loss) for math and coding models during finetuning, as well as the merged models. The math and coding models exhibit steady decreases in validation loss as they specialize on their respective tasks. In contrast, the validation loss of the merged model via activation interpolation on a cross-domain task requiring both math and coding decrease quickly and increase gradually after a critical point. (Middle) Merging loss plotted against CKA similarity computed on data from the adaptation dataset. (Right) Merging loss plotted against CKA similarity computed on data from the pretraining dataset.

In summary, representational compatibility appears to play a crucial role in the success of merging specialized models. As models specialize, their internal representations diverge, making alignment across models over time increasingly difficult. This divergence is particularly evident when merging models that are highly specialized. To improve the effectiveness of model merging in feature space, strategies that promote representational similarity, either through modifications in pretraining or fine-tuning process as well as architectural are essential. In the following section we explore more complex merging methods to see if our current merging practices are limiting performance.

# 3 MERGING WITH MORE DEGREES OF FREEDOM IMPROVES ADAPTATION PERFORMANCE

> Are there limits to more sophisticated routing strategies for merging?

## 3.1 ROUTING IS MORE EFFECTIVE THAN STATIC INTERPOLATION

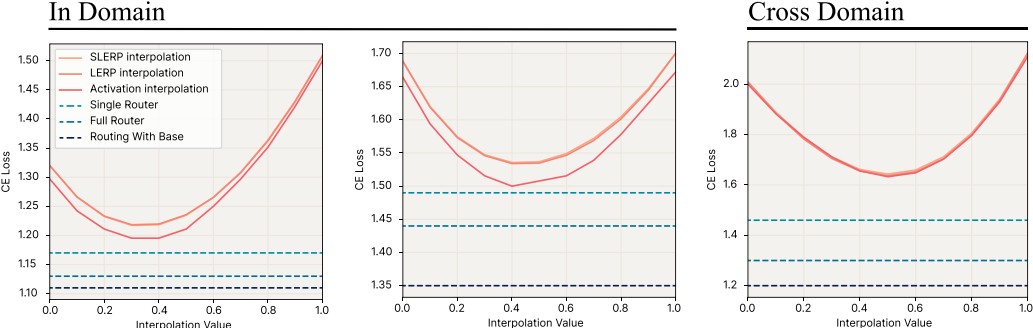

Figure 2: Performance comparison of various model merging techniques for In-Domain and Cross-Domain tasks. The plot shows the progression of different merging methods, from simple interpolation strategies, (SLERP, LERP, activation interpolation) see A.5, to more complex ones involving routers (Single Router, Full Router, Routing with Base Model). The trend demonstrates that increasing the complexity and capacity for model merging results in performance gains, as reflected by the lower adaptation loss.

We provide a characterization of different merging methods based on their **degrees of freedom** which describes how large their search space is. For example, when we do a simple interpolation of parameters where $\theta_{\text{merge}} = \alpha\theta_A + (1-\alpha)\theta_B$, we are searching along a one dimensional subspace. Searching the right subspace has shown to be important (Wortsman et al., 2021). Pushing this direction, we focus our attention on routing methods, where each router can explore an interpolation between two or more experts conditioned on the current input or token. By introducing routers we can dynamically combine information between layers from different models, thus increasing our search space for merging.

As shown in Figure 2, in all tasks as the complexity of the merging method increases in degrees of freedom, we observe an improvement in the adaptation performance. We see that merging scales with more routers and models. To push current

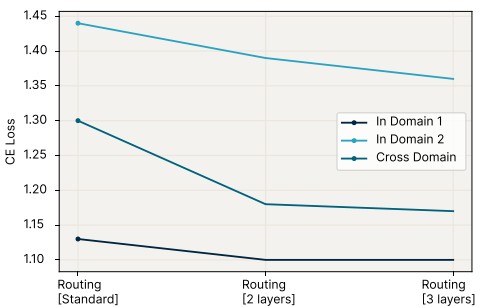

Figure 3: Comparison three routing strategies in model merging: *Standard*, *2-Layer*, and *3-Layer Routing* (see Figure 4). Evaluated on two in-domain tasks and one cross-domain task, results show that increased routing complexity reduces CE loss across all tasks. *2-Layer Routing* achieves notable gains over standard routing, with *3-Layer Routing* offering further, minor improvements.

methods we will describe in the following section how we increased the search space for routing based merging.

## 3.2 MULTI-LAYER ROUTING IMPROVES MERGING PERFORMANCE

Based on the aforementioned trend we further increase the search space by allowing the router at layer $l$ of the MoE to route to expert MLPs at different layers than $l$ across

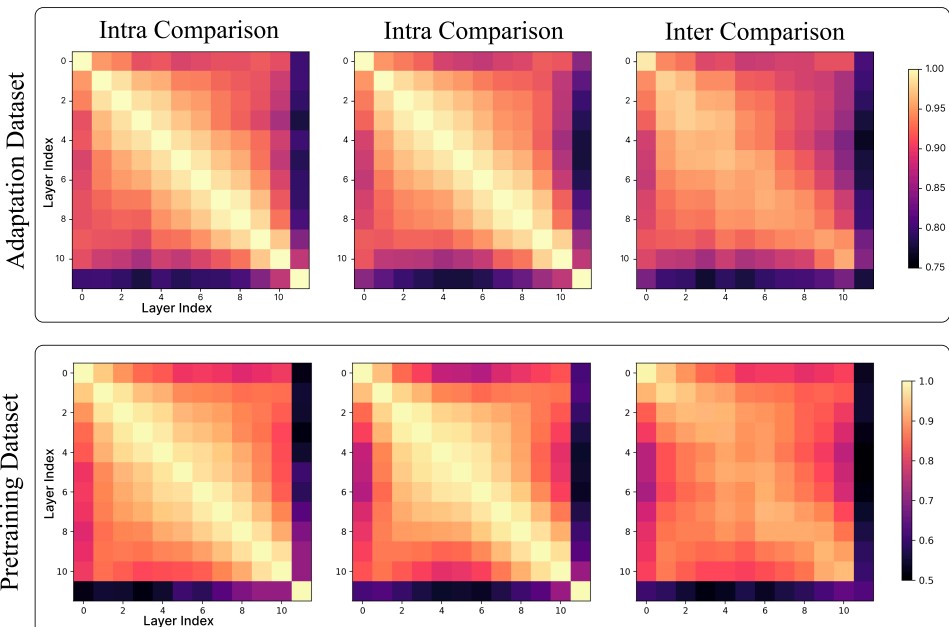

Figure 4: Visual representation of different routing strategies for model merging. From left to right: (1) *Standard Routing* performs layer-wise merging between corresponding layers from models A and B using a weighted average. (2) *Routing to 2 Layers* expands the merging process by incorporating not only the corresponding layers but also the next layer, allowing the router to combine outputs from the current layer and the layer above. (3) *Routing to 3 Layers* extends this further by merging the current layer, the layer above, and two layers above, enabling more complex

different models. Typically in an MoE model, experts will come from the same layer $l$ in the specialized models. This allows experts to be reused across layers and consequently it broadens the search space. We visually describe what this means in Figure 4.

Figure 5: CKA representational similarity analysis for MLP layers in a cross-domain math and coding task, comparing layer outputs across models. The top row shows comparisons based on the adaptation dataset, while the bottom row shows comparisons based on the pretraining dataset. The first two columns depict intra-model comparisons, illustrating the self-similarity of representations within the same model across different layers. These plots show that adjacent layers exhibit higher representational similarity, whereas layers farther apart have significantly lower similarity. The third column shows inter-model comparisons, reflecting the similarity of corresponding layers between the math and coding models. Layers in distant positions, demonstrate lower representational alignment.

However, there is a plateau in performance across all tasks as shown in Figure 3 when increasing the number of layers the router can route to. This implies that unless there exists a better way to efficiently decompose the model, the current method of MoE routing style merging is limited. Ideally, we would want to be able to create a more complex routing scheme that uses different blocks across different locations across different models, instead of being locally limited. In the following section we investigate this phenomenon through a representational lens.

### 3.3 THE CHALLENGES OF LAYER WISE REPRESENTATIONAL INCOMPATIBILITY

When the performance in Figure 3 plateaus, we find that is is correlated with representational similarity between layers within a model and across models. We specifically analyzed this in the cross domain experiment as shown in Figure 5. We pass a batch from the adaptation dataset and the pretraining dataset into both the math and coding finetuned models, and measure the representational similarity between different layers. It is apparent that within a model (*Intra*), layers adjacent to each other produce the most similar representations. In addition, even across the finetuned models (*Inter*) there is highest representational similarity around similar relative positions in the network. This suggests that layers with high representational dissimilarity can not have their outputs combined in a straightforward way.

Consequently, we are limited to routing experts that come from similar relative positions in the network. To mitigate this one would have to change the architecture, or how the models are pretrained or fintuned to ensure representationally compatibility across different layers in the network.

## 4 CURRENT LIMITATIONS AND FUTURE DIRECTIONS

Our experiments demonstrate that while routing-based merging shows promise, it still underperforms direct fine-tuning across all evaluated tasks (Table 1). This limitation necessitates fundamental improvements before model merging can be practically adopted. We outline key challenges and proposed solutions toward achieving decentralized collective intelligence (Raffel, 2023).

| Method | Cross-Domain | In-Domain 1 | In-Domain 2 |
|---|---|---|---|
| Merging | 1.17 | 1.10 | 1.36 |
| Fine-tuning | **1.04** | **0.91** | **1.01** |

Table 1: Loss comparison of merging and fine-tuning methods on the cross-domain task.

### 4.1 REPRESENTATIONAL COMPATIBILITY CHALLENGES

While successful cases of model combination exist, such as LLaVA's vision-language model stitching (Liu et al., 2024; Li et al., 2024), these typically involve carefully designed adapters between two models. Scaling this approach to dynamic routing among multiple models would require exponentially many adapters. Despite progress in layer-wise compatibility (Jiang et al., 2024b), ensuring cross-model representational consistency remains an open challenge.

### 4.2 INPUT-OUTPUT SPACE ROUTING

Drawing parallels to open-source software development (Raffel, 2023), we propose that one fruitful direction of research is to shift from feature-space merging to routing models in their input-output spaces. Rather than attempting to merge internal representations, this approach treats models as specialized functions operating in a common space (e.g., language), similar to how software libraries are composed.

### 4.3 ROUTER DESIGN CONSIDERATIONS

Current MoE routing mechanisms are constrained by fixed expert counts and requirements for smooth loss interpolation. Future routers should mirror how programmers select libraries: actively searching and retrieving models based on task requirements. This suggests an RL-based approach to routing, combined with clear model functionality descriptions (Lee et al., 2024). Such descriptions could include human-provided specifications and input/output examples when models are contributed to a collective repository.

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

## A  APPENDIX

We present the main tools we will use. In sub-section A.1, we describe the general Mixture of Experts formulation and how we employ it for model merging. In sub-section A.2, we outline the merging methods we compare. In sub-section A.3, we discuss the tasks used to evaluate the merging methods. Finally, in sub-section A.4, we explain how we analyze model representations and measure similarity. Then we will present more ablations and analysis.

### A.1  MIXTURE OF EXPERTS FOR MERGING

In our work, we investigate model merging based on the Mixture of Experts (MoE) framework (Yadav et al., 2024a). Instead of typically merging models into a single compressed one, we maintain the original models and introduce a router that dynamically weights token features outputted by different experts. Typically, the MLP layer in a transformer (Vaswani, 2017) is treated as the expert in an MoE (Fedus et al., 2022).

Let $\{E_i\}_{i=1}^N$ denote $N$ experts. For a sequence of $T$ tokens in $\mathbb{R}^d$, represented by $\boldsymbol{X} \in \mathbb{R}^{T \times d}$, the router $R(x) : \mathbb{R}^d \to \Delta_N$ assigns a distribution over the experts for each token $x_t \in \mathbb{R}^d$.

The output of the MoE layer for token $x_t$ is a weighted sum of expert outputs:

$$y_t = \sum_{i=1}^N r_{t,i} \, E_i(x_t),$$

where $r_{t,i}$ is the weight assigned by the router to expert $E_i$ [1]. Thus, for the entire sequence $\boldsymbol{X}$, the output is $\{y_t\}_{t=1}^T$.

In our work, the routing function $R(x_t)$ is a linear transformation. Given an input token $x_t \in \mathbb{R}^d$, the router is parameterized by a weight matrix $\boldsymbol{W} \in \mathbb{R}^{N \times d}$. The router output $r_t$ is computed as:

$$r_t = \mathrm{softmax}(\boldsymbol{W} x_t),$$

We now describe how we utilize the MoE formulation for model merging. During fine-tuning, we train only the MLP layers to create specialized models. The MLP layers become our "experts". Additionally, we train the router exclusively on the adaptation dataset. We capture this in Figure 6. Performance is evaluated through auto-regressive next token prediction on a held-out validation set, and we report results using cross-entropy loss (CE Loss).

### A.2  DIFFERENT MERGING METHODS

While merging methods typically are a form of parameter averaging, we find that MoE routing style merging performs consistently better and shift our focus on them. We now describe the different routing schemes we investigate.

- **Activation Interpolation** Instead of interpolating the experts via a router, we have a static coefficient $\alpha \in [0, 1]$.
  So for layer $l$, $x^{(l)} = \alpha \, E_A^{(l)}(x^{(l-1)}) + (1 - \alpha) \, E_B^{(l)}(x^{(l-1)})$.
- **Single Router** We place a single trainable router after the first attention layer to produce the routing coefficients $r_t$ that are used for weighted averaging of the MLP outputs across all layers.
- **Full Layer Routing** Instead of just having a single router determining the weighting between MLP outputs, we have a separate routing function per layer, more similar to a standard MoE.

---

[1]Note that the router can be sparse and only utilize a subset of experts as well

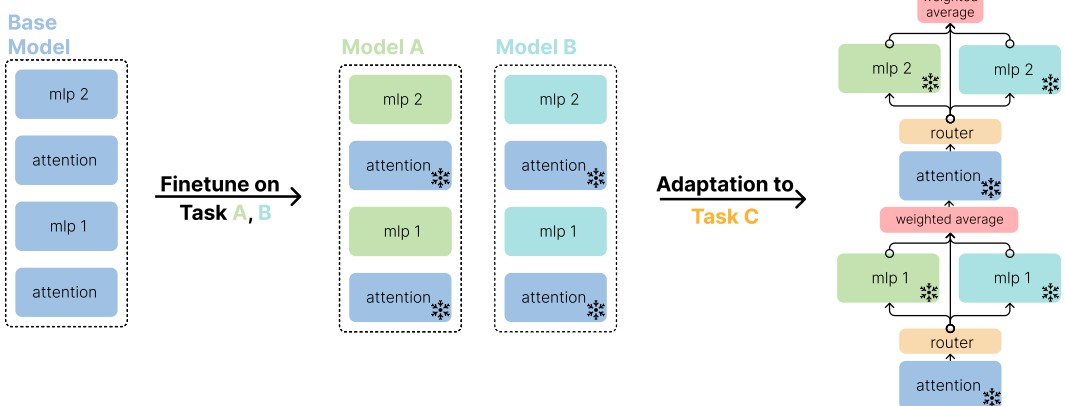

Figure 6: We illustrate the general routing based merging pipeline as follows. First, the MLP layers are finetuned from a base model on specialized datasets. Once we obtain a set of specialized models, we construct a Mixture of Experts (MoE) where the experts are the finetuned MLP layers from the various models. Finally, on a novel adaptation dataset, we train only the router.

- **Full-Layer Routing with Base Model** In our experiments we obtain specialized models by fine-tuning MLP layers from a common base model. Instead of only routing between the finetuned models, we also include the base model as well as another model that can be merged. This tests how routing scales with access to more models.

- **Multi-Layer Routing** We extend MoE merging by allowing the router at layer $l$ to route to experts from different layers. This allows for reuse of experts and more complex routing paths.

## A.3 TASKS

We now describe the different tasks we evaluate on.

- **In-Domain** We fine-tune a pretrained GPT-2 (Radford et al., 2019) model using the nanoGPT codebase on two different HuggingFace coding datasets[2]. In all fine-tuning, we only update the MLP layers and freeze the attention layers to allow for easy MoE-style routing for merging. We evaluate merging adaptation on **two** additional coding datasets[3].

- **Cross-Domain** We investigate how well merging can be used for cross-domain adaptation. To this end, we fine-tune a pretrained GPT-2 model on a math dataset[4] and a coding dataset[5]. We measure merging adaptation on a dataset[6] that requires both math and coding reasoning.

We report results in validation Cross-Entropy (CE) Loss.

## A.4 CENTERED KERNEL ALIGNMENT

We will be utilizing the centered kernel alignment (CKA; Kornblith et al., 2019) metric to compare the representations of two models. CKA is a well established metric that has been used in numerous analysis works (Raghu et al., 2021; Lange et al., 2022; Phang et al., 2021).

---

[2] `nampdn-ai/tiny-codes` and `TokenBender/code_instructions_122k_alpaca_style`

[3] `nickrosh/Evol-Instruct-Code-80k-v1` and `open-phi/programming_books_llama`.

[4] `microsoft/orca-math-word-problems-200k`.

[5] `nampdn-ai/tiny-codes`.

[6] `reasoning-machines/gsm-hard`.

It is important to note that the CKA metric is invariant to isotropic scaling, biases, and orthogonal transformations to the representations.

Given two sets of representations, $\mathbf{X} \in \mathbb{R}^{n \times d_1}$ and $\mathbf{Y} \in \mathbb{R}^{n \times d_2}$.

$$\mathbf{H} = \mathbf{I} - \frac{1}{n}\mathbf{1}\mathbf{1}^{\top}, \quad \overline{\mathbf{K}} = \mathbf{H}\mathbf{K}\mathbf{H}, \quad \overline{\mathbf{L}} = \mathbf{H}\mathbf{L}\mathbf{H}$$

$$\text{HSIC}(\mathbf{K}, \mathbf{L}) = \frac{1}{(n-1)^2} \text{tr}(\overline{\mathbf{K}}\,\overline{\mathbf{L}})$$

$$\text{CKA}(\mathbf{X}, \mathbf{Y}) = \frac{\text{HSIC}(\mathbf{K}, \mathbf{L})}{\sqrt{\text{HSIC}(\mathbf{K}, \mathbf{K}) \cdot \text{HSIC}(\mathbf{L}, \mathbf{L})}}$$

where $\mathbf{H}$ is a centering matrix, and $\mathbf{K}_{i,j} = K(x_i, x_j)$ and $\mathbf{L}_{i,j} = L(y_i, y_j)$ are kernel matrices generated by the kernel functions $K, L$. We use the dot product as the kernel function in our experiments. $\text{HSIC}(\cdot, \cdot)$ is the empirical estimator for the Hilbert-Schmidt Independence Criterion (Gretton et al., 2007), which was originally developed to measure the statistical independence of random variables.

## A.5 Weight Interpolation Methods

**Linear Interpolation (LERP)** Given two models' weights $\theta_A, \theta_B$, we search along the one dimensional subspace: $\alpha\theta_A + (1-\alpha)\theta_B$, $\alpha \in [0, 1]$.

**Spherical Interpolation (SLERP)** One potential issue with linear interpolation is that if the normalized dot product between two flattened weights is close to $-1$, then a linear interpolation can result in a low norm weight. To mitigate this effect, we perform spherical interpolation of the weights, where we search along the arc. Let $\mathbf{v}_0, \mathbf{v}_1$ be two flattened weight vectors, we switch weight notation from $\theta$ to $\mathbf{v}$ avoid confusion when referring to angles between weights.

$$\text{slerp}(\alpha, \mathbf{v}_0, \mathbf{v}_1) = \frac{\sin((1-t)\theta)}{\sin(\theta)}\mathbf{v}_0 + \frac{\sin(t\theta)}{\sin(\theta)}\mathbf{v}_1$$

where $\theta = \cos^{-1}(\mathbf{v}_0^{\top}\mathbf{v}_1)$. If the vectors are nearly colinear (i.e., the dot product is close to 1), linear interpolation (LERP) is used:

## A.6 Architecture Details

| params | dimension | $n$ heads | $n$ layers | learning rate | batch size | $n$ tokens |
|--------|-----------|-----------|------------|---------------|------------|------------|
| 124M | 768 | 12 | 12 | $8.0 \times 10^{-4}$ | 64 | 160k |

Table 2: Model sizes, architectures, and optimization hyper-parameters for fine-tuning GPT2

We use a learning rate of $8 \times 10^{-5}$ for the fine-tuning results in Table 1

## A.7 In Domain Layer-Wise Similarity

See Figure 7 for the layer wise representational analysis between two coding models across two different coding adaptation tasks.

## A.8 Routing Ablations

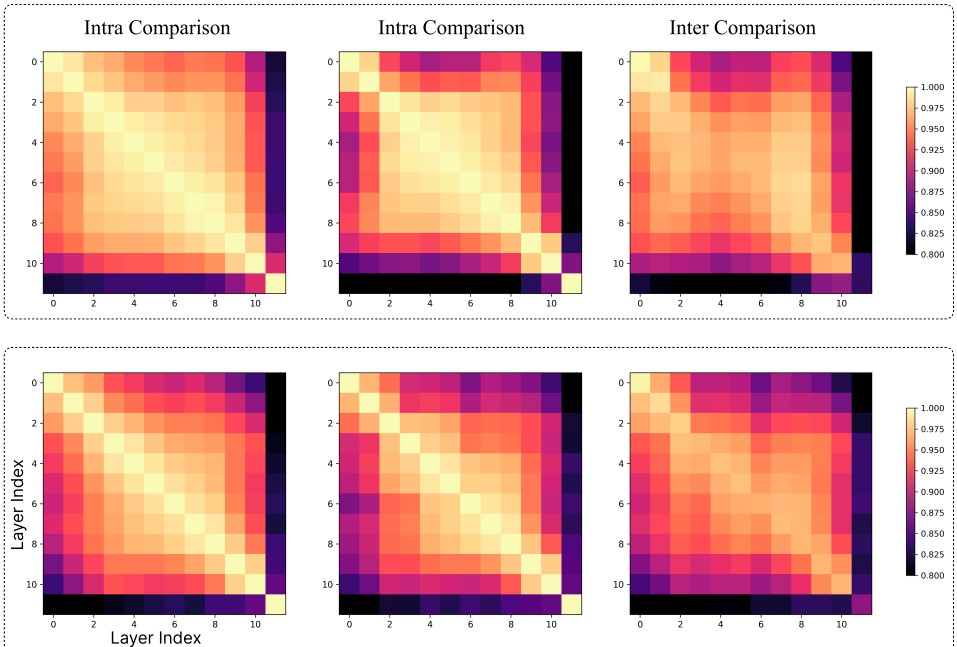

Figure 7: In the in-domain tasks, we plot the CKA representational distance between different MLP layer outputs for the same batch of inputs. The first two columns show the self-similarity of representations within the same model at different layers, illustrating how adjacent layers are more representationally similar. The right column shows the similarity between the representations of corresponding layers from two different models.

|  | Cross-Domain CE Loss |
|---|---|
| DataMix Routing | 1.35 |
| 2 Layer MLP Router | 1.28 |
| Standard Routing | 1.30 |

Table 3: Cross Domain Performance with different ablations. DataMix Routing refers to routing a math and coding policy but the math and coding models were co-finetuned on the pretraining data. 2 Layer MLP refers to a router being a 2 layer MLP to test the effects of having more routing capcity.

