# OpenReview forum: "Collective Model Intelligence Requires Compatible Specialization"
_ICLR.cc/2025/Workshop/MCDC — MCDC @ ICLR 2025_

### Official Review · Reviewer_P1KA · 2025-02-26

**Rating:** 6
**Confidence:** 3
**Fit:** 4

**Summary:**

This paper explores the limitations of model merging through parameter and feature averaging, identifying representational divergence as a major barrier to achieving collective model intelligence. The authors introduce the concept of compatible specialization, arguing that models must not only be specialized but also maintain compatibility in feature representations to be effectively merged. Using centered kernel alignment (CKA), they show there's a critical point during fine-tuning where model representations become too divergent for effective merging.

The authors explore routing-based merging strategies as a more flexible alternative to simple parameter averaging. They test several approaches with increasing degrees of freedom, from simple interpolation strategies to more complex routing methods. While more complex routing strategies demonstrate improved performance, they still underperform direct fine-tuning. The authors suggest shifting from feature-space merging to input-output space routing, similar to how software libraries are composed, to achieve true collective model intelligence.

**Reason For Giving A Higher Score:**

The paper provides a pretty good theoretical and empirical foundation for "compatible specialization" in model merging. The use of CKA analysis, the identification of a critical threshold for merging failure, and the insightful comparisons between interpolation-based and routing-based merging are valuable contributions. The novel extension of MoE routing enhances model flexibility, and the proposed future directions offer promising avenues for improving specialization compatibility.

**Reason For Giving A Lower Score:**

Most importantly, key methodological details are buried in the appendix, making the paper less readable. This paper requires major refinement and restructuring between the main text and the appendix.  The evaluation is limited in scope, focusing only on math and coding and only on GPT-2 architecture, raising concerns about generalizability. As the paper is not focused on the solution but explores the compatible specialization issue, I expect an expansion of the experiments to collect more evidence.

**Strengths And Weaknesses:**

**Strengths:**

- The paper introduces "compatible specialization" a key challenge in model merging, highlighting the trade-off between specialization and representation compatibility and its effect on the performance of model merging.

- The use of CKA to analyze representational similarity provides good evidence of why merging becomes difficult as models specialize. The identification of a critical threshold (t) where merging fails is valuable. The comparisons between interpolation-based and routing-based merging provide useful insights.

- The authors extend standard MoE routing to allow experts to be reused across different layers, which is a novel contribution. Results suggest that increasing routing complexity improves performance, though with diminishing returns.

- The paper suggests promising future directions for achieving compatible specialization through input-output space routing instead of feature-space merging.

**Weaknesses:**

- Limited Evaluation Scope: The evaluation is restricted to math and coding domains. It would be great to extend it to more diverse domains, languages, or even modalities to make a more generalizable conclusion. The definition of cross-domain tasks is unclear. It would be great to see the train and validation data examples.  It is not evident how merging performs on each task separately rather than in a combined setting. The study only considers GPT-2, raising concerns about generalizability to other architectures (e.g., LLaMA, T5, Mistral, etc.).

 - Unfair Comparison Between Routing and Interpolation: The MoE routing approach requires training with adaptation data, while interpolation-based methods do not, making the comparison potentially unfair. When comparing finetuning with routing and merging, it is always important to consider the trade-off between memory+computation cost and performance.

 - Missing Baseline Comparisons: The study only tests LERP, SLERP, and Activation interpolation, while several state-of-the-art (SoTA) merging techniques exist such as TIES-Merging, Dare-merging, DellaMerging, or Evolutionary Model Merging (parameter + data-space merging). Without these baselines, it is difficult to assess whether routing is better than model merging. Also, the performance of each merging technique is so sensitive to its parameter setting. It is not easy without the help of an appendix to understand the details.

- Lack of Clear Methodology: Understanding the finetuning details, pretraining or adaptation datasets, parameter settings, different routing, and merging techniques requires reading the appendix, whereas these details should be clear from the main text

**Suggestions:**

- Expand task diversity: Test on a broader range of tasks, languages, or domains to strengthen generalizability claims. Even, try more than two domains for multi-domain settings. This might be insightful when comparing routing versus interpolation approaches as load-balancing issues might appear in more diverse task settings.

 - Include larger models and more diverse model architectures: Evaluate whether the findings hold for other state-of-the-art models beyond GPT-2.

 - Compare with additional baselines: Include comparisons with other recent model merging techniques. Also, show how you selected the merging parameters

 - Must-Do: Refine the paper’s structure and figure orders. Some of the important information about the methodology is hidden in the Appendix and this significantly reduces the readability of this paper.

 - Nice-to-Have: Develop prototype solution: Implement at least a preliminary approach for compatibility-aware model merging or input-output routing to demonstrate feasibility.

 - Nice-to-Have: Extend to non-language models: Test whether the findings generalize to other model types like vision transformers or multimodal models.

 - Nice-to-Have: Analyze attention mechanisms: Investigate whether attention layers show similar compatibility issues as MLP layers.

**Minor Comments:**
 - Page 3, line 117: It is import -> It is important
 - Page 5, line 266: that is is -> that is

---

### Official Review · Reviewer_ZKPw · 2025-02-28

**Rating:** 7
**Confidence:** 4
**Fit:** 5

**Summary:**

This paper explores compatible specialization as a fundamental prerequisite for successfully merging models fine-tuned on distinct tasks. The authors argue that parameter or feature-based averaging is inherently limited due to increasing representational divergence as specialization progresses. Specifically, models fine-tuned on disparate domains (e.g., mathematics and programming) develop distinct internal feature structures, rendering direct interpolation suboptimal. To mitigate this, the authors propose a mixture-of-experts (MoE) framework that leverages routing mechanisms to dynamically select and integrate specialized layers. While this approach demonstrates improvements over simple averaging in controlled settings, it remains constrained by the extent of representational dissimilarity among specialized layers. Through empirical analysis on both in-domain and cross-domain tasks, as well as Centered Kernel Alignment (CKA) evaluations, the study illustrates the challenges of model mergeability and posits that future research should explore architectural innovations that facilitate structured model communication rather than relying on direct feature alignment.

This study contributes to the growing discourse on model fusion, highlighting the limitations of traditional parameter aggregation techniques and advocating for alternative strategies that emphasize structured knowledge transfer. The findings suggest that future advancements in model merging should prioritize compatibility mechanisms over naive averaging methods, leveraging explicit pathways for model interaction.

**Reason For Giving A Higher Score:**

- Timely Topic: Model merging and collaborative intelligence are active areas of research, and the paper addresses a real challenge: how to combine specialized models effectively.
- Useful Empirical Findings: The experiments demonstrate that naive feature interpolation has inherent limitations, and routing improves performance in several cases. These insights could spark new ideas for future model-merging methods.

**Reason For Giving A Lower Score:**

- Lack of Robust Formalism: While “compatible specialization” is introduced, it remains somewhat abstract. A more rigorous framework or metric for what compatibility entails would strengthen the paper’s impact.
- Limited Exploration of Alternatives: The paper largely compares routing methods to basic averaging without broader baselines, making it hard to position this work among other existing or emerging techniques for merging or modularizing specialized models.
- Performance Gap: Even with routing, the merged models typically underperform standard fine-tuning for new tasks. That limitation, though well-disclosed, might temper excitement about practical utility in real-world multi-task scenarios.

**Strengths And Weaknesses:**

### Strengths
- Relevant Topic: The paper addresses a key challenge in model merging, a growing concern as pre-trained models become more widespread.
- Clear Motivation: The study convincingly argues for routing-based merging over naive feature interpolation.
- Empirical Support: The experiments demonstrate that simple feature averaging is insufficient for specialized models, reinforcing the need for improved merging techniques.

### Weaknesses
- Abstract Definition of Compatibility: The concept of compatible specialization is important but lacks precise formalization.
- Limited Baseline Comparisons: The paper does not benchmark against more advanced merging methods, reducing its comparative insight.
- Performance Trade-offs: Even with routing, the merged models do not always outperform fine-tuning, highlighting unresolved challenges.

**Suggestions:**

- Discuss Specialization vs. Mergeability Trade-offs: The paper underscores the challenge of merging highly specialized models, but further discussion on the trade-offs between preserving specialization and ensuring mergeability would be valuable. Addressing whether these trade-offs can be systematically optimized would be particularly beneficial.
- Expand Baseline Comparisons: The current baselines primarily focus on feature and parameter averaging. Comparing the proposed approach against alternative merging techniques such as linear mode connectivity or ensemble distillation would better contextualize its performance.
- Analyze Computational Overheads: While routing provides more flexibility in merging, it introduces increased computational complexity. A discussion on the trade-offs between performance gains and efficiency costs would aid practitioners in evaluating the feasibility of routing-based merging.

---

### Official Review · Reviewer_1Loe · 2025-03-01

**Rating:** 5
**Confidence:** 3
**Fit:** 4

**Summary:**

This paper examines the difficulties of merging specialized models to achieve collective intelligence. It finds that simply averaging model features fails because specialized models develop incompatible internal representations. The authors analyze this divergence and test routing-based merging strategies but find that even these flexible methods have limitations. The paper highlights that effective model merging requires structured input-output communication rather than direct integration of neural activations. Experiments show that increased complexity in routing strategies does improve adaptation performance, but a plateau is eventually reached due to persistent representational incompatibility between layers. The authors suggest that future work should focus on enabling models to communicate more like humans---using language instead of merging internal representations---and propose approaches like RL-based routers and clear model descriptions to improve collaboration.

**Reason For Giving A Higher Score:**

See the strengths

**Reason For Giving A Lower Score:**

See the weaknesses

**Strengths And Weaknesses:**

Strengths:
- The paper highlights that current model merging methods face challenges in effectively combining specialized models because fine-tuning causes representational divergence. Using centered kernel alignment (CKA), the paper shows that as models become more specialized, their feature space similarity decreases.
-  The paper explores routing-based merging strategies as a more flexible approach to combining specialized models by dynamically routing across different layers. It shows that increasing the complexity and capacity of model merging through routing results in performance gains.
- The paper emphasizes the need for new model merging approaches that operate within well-defined input and output spaces, akin to human communication through language. This shift moves away from merging internal representations and instead focuses on enabling models to exchange information effectively.
- The paper provides insights for future work by advocating a shift toward routing models in their input-output spaces, treating them as specialized functions within a shared space. It also suggests key design considerations for routers, including an RL-based approach to routing and the use of clear model functionality descriptions.

Weaknesses:
- The paper discusses a trade-off between specialization and compatibility. It suggests that beyond a certain point, increased specialization can hinder a model's ability to merge effectively, resulting in diminishing returns. This suggests that achieving the right balance between specialization and the ability to merge models is a challenge that the paper highlights but does not fully resolve.
- The paper highlights the need for compatible specialization but offers no specific solution for achieving it. It primarily focuses on highlighting the shortcomings of current model merging techniques without providing a clear path forward or a practical implementation.
- The experiments primarily focus on specific tasks such as math, coding, and cross-domain adaptation using GPT-2 models. The generalizability of the findings to other tasks, models, and domains may be limited.

**Suggestions:**

- The authors could explore regularization techniques during fine-tuning to encourage models to maintain a degree of representational similarity.
- The authors propose shifting from feature-space merging to routing models in their input-output spaces. They could provide a more detailed exploration of how this could be implemented, potentially including a preliminary implementation or simulation.
- The authors should expand their experiments to cover a broader range of tasks and models, including larger models.

---

### Decision · Program_Chairs · 2025-03-06

**Decision:**

Accept

**Comment:**

This paper argues that specialized models develop incompatible internal representations leading to difficulties in merging specialized models to achieve collective intelligence. Most of the reviewers liked the paper, found it relevant to the workshop, and recommended acceptance. We suggest the authors to incorporate the comments of the reviewers to further strengthen the paper. Overall, we're recommend to accept this work to the workshop.